cognition/neuroscience/behaviour

peripersonal space, group membership, audio-tactile integration, collaborative behaviour, looming sound, auditory space

**Author for correspondence:**
Lise Hobeika
e-mail: lise.hobeika@gmail.com

# Social coding of the multisensory space around us

Lise Hobeika[1], Marine Taffou[1,2]
and Isabelle Viaud-Delmon[1]

[1]CNRS, Ircam, Sorbonne Université, Ministère de la Culture, Sciences et Technologies de la Musique et du son, STMS, F-75004 Paris, France
[2]Institut de Recherche Biomédicale des Armées, 91220 Brétigny-sur-Orge, France

LH, 0000-0002-1257-7187

Multisensory integration of stimuli occurring in the area surrounding our bodies gives rise to the functional representation of peripersonal space (PPS). PPS extent is flexible according to the affective context and the target of an action, but little is known about how social context modulates it. We used an audiotactile interaction task to investigate PPS of individuals during social interaction. Participants had to detect as fast as possible a tactile stimulus while task-irrelevant looming sounds were presented, while they were paired as collaborative dyads and as competitive dyads. We also measured PPS in participants seated near an inactive individual. PPS boundaries were modulated only when participants collaborated with a partner, in the form of an extension on the right hemispace and independently of the location of the partner. This suggests that space processing is modified during collaborative tasks. During collective actions, a supra-individual representation of the space of action could be at stake in order to adapt our individual motor control to an interaction as a group with the external world. Reassessing multisensory integration in the light of its potential social sensitivity might reveal that low-level mechanisms are modified by the need to interact with others.

## 1. Background

The space around the body, called peripersonal space (PPS), is the space where physical interactions between the body and the external world occur. PPS is opposed to the more distant extra-personal space. Neurophysiological studies in monkeys as well as brain imaging and lesion studies in humans have evidenced this dichotomy and showed that PPS is coded in the brain through multisensory integration mechanisms [1–6]. This dedicated multisensory integration system is thought to have a role in apprehending events occurring in the space near the body, which requires the implementation of particularly

appropriate and precise behaviours: be it for dealing with imminent threat [4] or for attaining an object of interest [7]. Furthermore, PPS displays flexible boundaries that are sensitive to motor action factors [8,9] as well as to affective context [10,11], so that multisensory integration related to the body can be adaptively boosted according to the situation for the purpose of actions.

A turn in neuroscience has been recently introduced with the investigation of the brain mechanisms underlying social and affective psychological processes, spurred on by the fact that human species is a social species and that most of humans' behaviours are dependent on their social environment [12]. Going beyond the investigation of social stimuli processing in isolated individuals, the study of human cognition in social contexts with real-time social interaction paradigms has revealed the possibility to map social behaviour to particular brain mechanisms [13]. Several studies exploring the social roots of perception and action have shown that the spatial coding of stimuli is modulated when two individuals are acting together [14,15]. For example, in a 'social Simon task' paradigm, two participants seated side by side performed a go/no go version of a spatial compatibility task. A spatial compatibility effect—known as Simon effect [16]—appeared only when participants performed the task as co-actors, as if they were dividing the motor space between them. This spatial compatibility effect disappears when the two co-actors are located too far from each other [17], if they are involved in a negative, competitive relationship [18], and when they are experimentally affiliated with two different arbitrary groups [19].

While PPS is closely linked to the notion of proxemics developed by Hall [20], and a series of studies has been conducted on the influence of the social context and group membership on interpersonal distances [21,22] and on the perception of reachable space [23,24], few studies have addressed the question of whether and how multisensory integration related to PPS is influenced by social factors. Heed *et al*. measured the effect of task sharing on visuo-tactile integration within PPS. They measured a crossmodal congruency effect (i.e. a spatial congruency acceleration effect [25]) as participants did a task alone or shared it with a partner located within their PPS. They observed that performing a task together with a partner modulates multisensory processes within PPS and proposed that this modulation may allow others' actions in the space near the body [26]. Teneggi *et al*. measured participants' PPS extent in presence of an inactive person, who had had a cooperative or egoist behaviour in an economical game performed before the measure. The study showed that, after the economical game, PPS boundaries extended towards the partner, allowing their integration to one's PPS, only if the social interaction had been satisfying [27]. Comparable results were found manipulating the moral assessment of a humanoid in virtual reality [28]. Overall, those studies argue that socio-affiliative contexts modulate PPS boundaries.

However, studies assessing the immediate effect of social interaction and group membership on the spatial definition of PPS as defined by multisensory integration properties are critically missing. Experiments reported in the literature measured the post effect of social situations on PPS boundaries on single individuals facing inactive partners [27,28], therefore the individual is taken as the unit of analysis for understanding the impact of social phenomena on the coding of PPS. Other studies measured multisensory integration strength during joint action but at only one spatial position [26,29], therefore PPS boundaries could not be estimated. In order to assess how social interaction shapes perception, the dynamic multisensory coding of the space of action should be examined in two individuals acting concurrently, manipulating their social-affiliative factors. We hypothesized that there must be dedicated multisensory integration mechanisms supporting social interaction, and that the dynamic coding of PPS should be imprinted by group mechanisms, that might be different from individual PPS coding mechanisms. Here, we tested if experimentally induced social relation between two individuals can modify the dynamic coding of their PPS. We used minimal groups and manipulated group membership by varying the instructions across three experiments.

In the first experiment, we tested whether group membership would influence dynamic multisensory coding of the space around the body (collaborative experiment) by instructing a dyad of participants that they were both doing the same task and collaborating towards getting the best result possible. The task, presented as collaborative, allowed participants to share a common goal, to act concurrently but without motor coordination or subtasks repartition. In the second experiment, we introduced a social competition manipulation to prevent the dyad of participants to form a group (competitive experiment). In this competitive context, participants acted concurrently and performed exactly the same task as in the collaborative context (they had to respond as fast as possible). The difference is that in this situation, participants do not share the intention but rather pursue the goal as individuals. In a third experiment, we tested if the mere presence of an inactive individual changes PPS extent (audience experiment).

In previous experiments on the social coding of PPS [26–28], participants were facing an unknown individual and PPS extent was measured in the front space of participants. Yet, this configuration

prevents study of the link between the direction of the social modulation of PPS and the spatial position of the unknown individual, since the latter is co-localized with the measured PPS. Thus, it is not possible to conclude if PPS modulations are in the direction of the unknown individual, or if it is a global modulation of PPS in every direction. In the present series of experiments, we placed participants side by side, and measured PPS extent on the left and right hemispaces of participants. If PPS representation is sensitive to the social context, its modulation may be depending on the unknown individual's location, in the form of a contraction or an expansion that may be restrained to the hemispace where the unknown individual is located. However, a previous study reported that lateral PPS is asymmetrical for right-handers when tested in isolation, with a larger extent in the left than the right hemispace [30], which is consistent with a protective/defensive function of PPS. Right-handers would have a larger protective margin in the hemispace of their non-dominant hand [31]. Therefore, the presence of an unknown individual may impact PPS representation differently in the left and right hemispaces, due to the dominant hand of the participant and to the relative location of the unknown individual.

In the present experiments, participants performed a speeded tactile detection task while irrelevant sounds were looming towards them from the frontal hemifield, either from the left or the right hemispace. Tactile stimuli were delivered at different delays from sound onset (T1, T2, T3, T4, T5, T6, T7, T8, T9). As such, participants perceived the sound source at different distances from their body when they processed the tactile stimulus (from the farthest distance at T1 to the closest distance at T9). As the delay increased, the looming sound was perceived as closer. It is now well established that a sound speeds up tactile detection when located near, but not far from, the body, i.e. within PPS [32,33]. Consequently, we sought for the distance from participants' body, at which sound starts to boost tactile reaction times (RTs) as a proxy of PPS boundaries. In order to measure PPS of participants' trunk, participants were instructed to keep their hands aligned with their mid-sagittal plane and in contact to their body [34]. With this instruction, the paradigm aimed at measuring PPS of participants' trunk [34]. However, we cannot exclude the possibility that our measure is somehow associated to the peri-hand PPS of the responding hand. Given that lateral PPS is asymmetric for right-handers [30], we analysed separately the social effect on PPS boundaries on each hemispace. In order to test whether potential PPS social modulations were in the direction of the unknown individual, we included the factor PARTNER POSITION in the analysis.

# 2. Material and methods

## 2.1. Apparatus

We used a modified version of Canzoneri *et al.*'s audiotactile interaction task [33]. Participants sat on one of two chairs placed side by side (figure 1) in a soundproofed room. Both of their hands were palms-down on a table, aligned and in contact with their mid-sagittal plane. To control for the visual stimulation and gaze direction, participants were instructed to fix a permanent visual target located at 65 cm in front of them. Auditory stimuli were presented through Beyer Dynamic DT770 headphones. The auditory stimulus was a sound of bubbling water (32 bits, 44 100 Hz digitization), processed through binaural rendering using non-individual head related transfer functions (HRTF) of the LISTEN HRTF database (http://recherche.ircam.fr/equipes/salles/listen/). With this procedure, the virtual sound source location can be manipulated by rendering accurate auditory cues such as frequency spectrum, intensity, and inter-aural differences.

In order to simulate the looming source, the sound was processed in the Max/MSP(6.1.8) environment using the Spat library [35]. The simulation consisted of a sound source moving in a virtual shoebox room (650 m$^3$), in which first reflections up to order 3, and late reverberation were rendered dynamically. The spatialization of the direct sound and of the first reflections was rendered using non-individual head related transfer functions (HRTF) taken from the LISTEN HRTF database (http://recherche.ircam.fr/equipes/salles/listen/). In order to enhance the looming effect of the direct sound, binaural nearfield correction filters were used [36]. According to the direction of incidence, the sound pressure level delivered to the ears of the participant was ranging from 35 to 61 dBA and from 37 to 78 dBA, for the contralateral and the ipsilateral ears, respectively. None of the participants reported mislocalization or lack of externalization, and they all confirmed that they could hear clearly sounds from the left and right hemispaces.

The absolute distance estimation of sound sources is usually a difficult task for non-familiar stimuli in absence of reference, in which participants are not accurate [37,38]. However, participants are accurate in

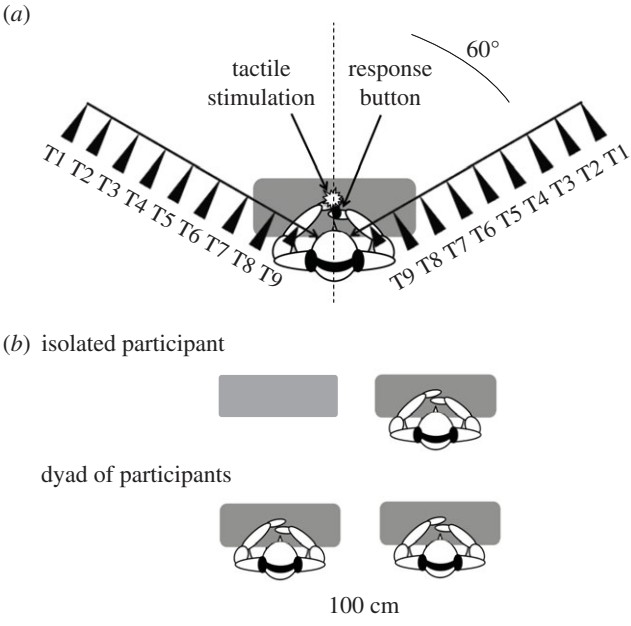

**Figure 1.** Experimental paradigm. (*a*) Participants performed the audiotactile task by responding to a tactile stimulation while a task-irrelevant sound was looming towards them from the left or right hemispace. On each trial, one tactile stimulation was delivered at one among nine possible delays from sound onset (T1 to T9), corresponding to nine possible distances of the sound source from participants' body. (*b*) Participants performed the task alone or in a dyad, seated side by side.

the comparison of distances between two sources at different distances. In our task, we used looming sounds, and compared the effect of continuously varying sound distances. The theoretical range of distance used in the experiments is from 135 to 20 cm to the head of the participant. As distance perception in depth is accurate for relative comparison but not in term of absolute distance, we will not present the results in terms of metrical distances but in terms of delay from the sound onset.

The tactile stimulus was a vibratory stimulus delivered by means of a 28 mm miniature speaker on the palmar surface of the left index finger of participants. A sinusoid signal was displayed for 20 ms at 250 Hz. With these parameters, the vibration of the loudspeaker was perceivable, but the sound was inaudible. A PC running Presentation® software was used to control the presentation of the stimuli and to record the responses.

## 2.2. Design and procedure

Participants were asked to place their left index finger on the vibrator and to press a button with their right finger each time a tactile stimulus was detected. A black fabric hid participants' hands. An auditory stimulus was presented for 3000 ms for each trial. The sound source approached from the front hemifield, either from the right (60°) or from the left hemispace (−60°). The auditory stimulus was preceded by 1000 ms of silence. A period of silence, with a duration varying between 2700 and 3300 ms, also occurred after the offset of the sound.

In 91.7% of the trials, a tactile stimulus was presented along with the auditory stimuli. The remaining 8.3% trials were catch trials with auditory stimulation only. Participants were instructed to ignore the auditory stimuli and to respond as quickly as possible to the tactile stimuli. They were asked to emphasize speed, but to refrain from anticipating. RTs were measured.

Temporal delays for the tactile stimulus were set as follows: T1 was a tactile stimulation administered simultaneously with the sound onset; T2 at 375 ms from sound onset; T3 at 750 ms from sound onset; T4 at 1125 ms from sound onset; T5 at 1500 ms from sound onset, T6 at 1875 ms from sound onset; T7 at 2250 ms from sound onset; T8 at 2625 ms from sound onset and T9 at 3000 ms from sound onset. Thus, tactile stimulation occurred when the sound source was perceived at different locations with respect to the body, i.e. far from the body at low temporal delays and close to the body at high temporal delays. Moreover, in order to measure RTs in the unimodal tactile condition (without any sound), tactile stimulation was also delivered during the silent periods, preceding or following sound administration, namely at −650 ms (Tbefore) and at 3650 ms (Tafter) from sound onset.

The total test consisted of a random combination of 10 target stimuli in each of the 44 conditions. There were a total of 440 trials with a tactile target, randomly intermingled with 40 catch trials. Trials were equally divided in 10 blocks of 48 trials, lasting about 5 min each. Participants were seated side by side, thus at the left or the right side of their partner. The within factors were: SOCIAL CONDITION (two levels: isolated/in dyad), HEMISPACE (two levels: left/right) and DISTANCE (eleven levels: Tactile_Before, T1–T9 and Tactile_After). The between subjects factor was the PARTNER POSITION (two levels: left/right).

## 2.3. Social manipulation

Three different experiments were conducted: the collaborative experiment, the competitive experiment and the audience experiment. In each experiment, every participant performed the audiotactile task in a main condition and in a control condition: respectively paired with another participant (dyad) and isolated. The order of the presentation of the two social conditions was counterbalanced between participants. In the dyad condition, participants were always paired with another participant of the same gender. The two participants were seated side by side, at 1 m from each other. In the isolated condition, an empty chair remained beside the participant. Each participant was assigned to the left or right seat and stayed in that position during the whole experiment.

The isolated condition was identical in every experiment but the social context in the dyad condition differed between the three experiments. In the collaborative experiment, the dyad of participants performed the audiotactile task as a team. We affiliated participants and created a group by giving them a shared goal. We told them that we were recording their global performance as a dyad and not their individual performance. In the competitive experiment, the dyad of participants performed the audiotactile task in competition. We told them that at each trial, we were recording only the fastest answer and not their individual performance. In the audience experiment, only one of the two participants from the dyad performed the audiotactile task while the other one was asked to stay inactive and immobile. This context, in which participants had no common task, was clearly explained to the participants in the instructions.

## 2.4. Participants

Eighty-eight healthy individuals with normal audition and touch took part in the study, either in the collaborative, the competitive or the audience experiment. Participants considered as outliers in terms of percentage of misses or global RTs were excluded from the analysis (see details in electronic supplementary material). The analyses were performed on the 81 remaining participants (collaborative experiment: $n = 26$ participants, 14 females, age m ± s.d.: 24.08 ± 4.39; competitive experiment: $n = 29$ participants 16 females, age m ± s.d.: 23.45 ± 4.15; audience experiment: $n = 26$ participants, 11 females, age m ± s.d.: 25.27 ± 3.58). All of them were right-handed. Participants received a financial compensation of 10€/hour for their participation. They provided a written informed consent prior to the experiment, which was approved by the Institutional Review Board of the French National Institute of Health and Medical Research (INSERM, IRB00003888).

# 3. Results

For each experiment, we first verified that tactile detection was speeded by simultaneous presentation of sound by comparing RTs in the bimodal and the unimodal conditions. In every experiment, this was verified for bimodals delays from T2 to T9, but not for T1 (at which tactile stimulation occurred at sound onset). Thus, RTs at T1 were excluded from the rest of the analysis (see details in electronic supplementary material). Then, we verified that PPS was anisotropic in the left and right hemispaces, for each experiment separately (see details in electronic supplementary material). After those verifications, we analysed the impact of the social condition on the PPS boundaries, separately on the left and right hemispaces.

## 3.1. Experiment 1: collaboration

In this experiment, we tested the impact of collaboration between two individuals on the coding of their PPS. Participants performed the audiotactile task either isolated or in dyad with a collaborative

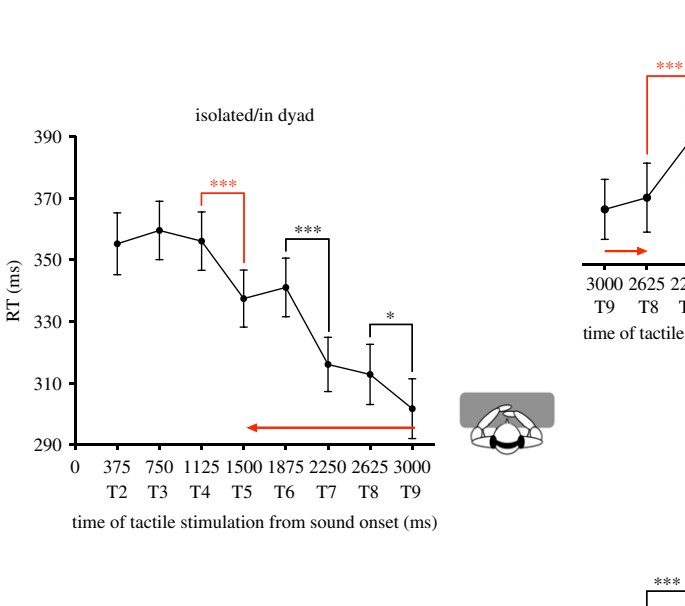

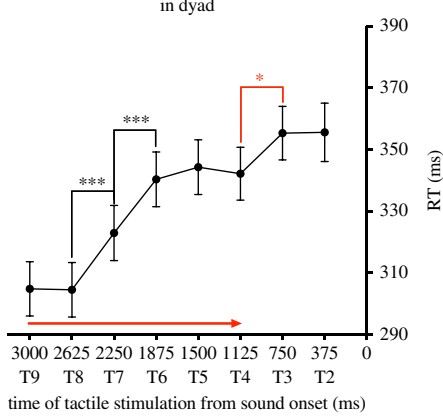

**Figure 2.** Collaborative context increases PPS extent in the right hemispace. This figure reports the mean tactile reaction times (±s.e.m.) as a function of the delay of tactile stimulation delivery from sound onset for the left hemispace (left graph) and for the right hemispace (right graphs). On the left hemispace, there was no effect of the social condition on RTs, thus participants' data in the condition isolated and in dyad are pooled. In the right hemispace, the social condition impacted PPS extent, thus RTs are presented separately for isolated participants (top graph) and participants acting in dyad (bottom graph). The significant decreases of RTs are indicated by means of asterisks. The significant decrease of RTs corresponding to the critical delay at which sound starts to boost tactile processing is indicated by means of red asterisks. The red arrows illustrate the extent of PPS boundaries from participants' body. In the right hemispace, PPS is larger when participants are acting in dyad than when they are doing the task alone.

instruction. We conducted separated ANOVA for the left and the right hemispace, with the between subject factor PARTNER POSITION (two levels: Left/Right) and the within subject factors SOCIAL CONDITION (two levels: Isolated/In Dyad), and DELAY (eight levels: T2, T3, T4, T5, T6, T7, T8, T9).

In the left hemispace, analysis indicated a significant effect of DELAY ($F_{7,168} = 36.77$, $p < 0.001$, $\eta_{\mathrm{p}}^2 = 0.605$). The other factors and interactions were not significant. Sound in the left hemispace started to boost tactile detection when tactile stimulation was administrated at T4 (figure 2). Participants' RTs were significantly faster when the tactile stimulus occurred at T5 compared to when the tactile stimulus occurred at T4 (*post hoc* Newman–Keuls' test: $p < 0.001$). RTs further decreased at the later delays. Participants' RTs were significantly faster when the tactile stimulus occurred at T7 than at T6 (*post hoc* Newman–Keuls' test: $p < 0.001$) and RTs were also significantly faster when the tactile stimulus occurred at T9 than at T8 (*post hoc* Newman–Keuls' test: $p < 0.05$). Moreover, RTs were significantly faster when the tactile stimulus was delivered at T5, T6, T7, T8 and T9 as compared to when the tactile stimulus was delivered at T2, T3 and T4 (*post hoc* Newman–Keuls' test: $p < 0.01$ in all cases). There was no significant effect of the factor PARTNER POSITION.

In the right hemispace, analysis revealed a significant two-way interaction SOCIAL CONDITION * DELAY ($F_{7,168} = 2.24$, $p = 0.033$, $\eta_{\mathrm{p}}^2 = 0.085$), and a significant main effect of DELAY ($F_{7,168} = 27.13$, $p < 0.001$, $\eta_{\mathrm{p}}^2 = 0.531$). The effect of the other factors and interactions were not significant. When

participants performed the task in a dyad, sound in the right hemispace started to boost tactile detection when tactile stimulation was administered at T4 (figure 2). Participants' RTs were significantly faster when the tactile stimulus occurred at T4 compared to when the tactile stimulus occurred at T3 (post hoc Newman–Keuls' test: $p = 0.016$). RTs further decreased at the later delays. Participants' RTs were significantly faster when the tactile stimulus occurred at T7 than at T6 (post hoc Newman–Keuls' test: $p < 0.001$) and RTs were also significantly faster when the tactile stimulus occurred at T8 than at T7 (post hoc Newman–Keuls' test: $p < 0.001$). Moreover, RTs were significantly faster when the tactile stimulus was delivered at T4, T5, T6, T7, T8 and T9 as compared to when the tactile stimulus was delivered at T2 and T3 (post hoc Newman–Keuls' test: $p < 0.05$ in all cases), except for RTs at T4 and T6 which are just marginally different ($p = 0.074$). In contrast, when participants were isolated, sound in the right hemispace started to boost tactile detection when tactile stimulation was administered at T8. Participants' RTs were significantly faster when the tactile stimulus occurred at T8 compared to when the tactile stimulus occurred at T7 (post hoc Newman–Keuls' test: $p < 0.001$). Moreover, RTs were significantly faster when the tactile stimulus was delivered at T8 and T9 as compared to when the tactile stimulus was delivered at T2, T3, T4, T5, T6 and T7 (post hoc Newman–Keuls' test: $p < 0.001$ in all cases). There was no significant effect of the factor PARTNER POSITION.

Those results show that PPS right boundaries were farther away when participants were performing the task together with another individual than when they were isolated. No social modulation was observed on left PPS boundaries. The change in PPS boundaries' location did not depend on whether the partner was sitting on the left or on the right side of participants. This suggests that PPS extent is sensitive to a high-level social instruction of collaboration.

## 3.2. Experiment 2: competition

In this experiment, we tested the impact of competition between two individuals on the coding of their PPS. Participants performed the audiotactile task either isolated or in dyad with a competitive instruction. We conducted separated ANOVA for the left and the right hemispace, with the between subject factor PARTNER POSITION (two levels: Left/Right) and the within subject factors SOCIAL CONDITION (two levels: Isolated/In Dyad), and DELAY (eight levels: T2, T3, T4, T5, T6, T7, T8, T9).

In the left hemispace, analysis indicated a significant main effect of DELAY ($F_{7,189} = 49.25$, $p < 0.001$, $\eta_p^2 = 0.646$) (figure 3). Sound in the left hemispace started to boost tactile detection when tactile stimulation is administered at T5. Participants' RTs were significantly faster when the tactile stimulus occurred at T4 compared to when the tactile stimulus occurred at T5 (post hoc Newman–Keuls' test: $p < 0.01$). RTs further decreased at the later delays. Participants' RTs were significantly faster when the tactile stimulus occurred at T7 than at T6 (post hoc Newman–Keuls' test: $p < 0.001$) and RTs were also significantly faster when the tactile stimulus occurred at T8 than at T7 (post hoc Newman–Keuls' test: $p < 0.01$). Moreover, RTs were significantly faster when the tactile stimulus was delivered at T5, T6, T7, T8 and T9 as compared to when the tactile stimulus was delivered at T2, T3 and T4 (post hoc Newman–Keuls' test: $p < 0.01$ in all cases). Analysis also revealed a significant main effect of the SOCIAL CONDITION ($F_{1,27} = 25.60$, $p < 0.001$, $\eta_p^2 = 0.488$). Participants' RTs were significantly faster when participants performed the task in a dyad than alone (post hoc Newman–Keuls' test: $p < 0.001$) (figure 3b). There was no significant effect of the factor PARTNER POSITION.

In the right hemispace, analysis indicated a significant main effect of DELAY ($F_{7,189} = 56.56$, $p < 0.001$, $\eta_p^2 = 0.677$) (figure 3). Sound in the right hemispace started to boost tactile detection when tactile stimulation is administered at T7. Participants' RTs were significantly faster when the tactile stimulus occurred at T7 compared to when the tactile stimulus occurred at T6 (post hoc Newman–Keuls' test: $p < 0.001$). RTs further decreased at the later delays. Participants' RTs were significantly faster when the tactile stimulus occurred at T7 than at T6 (post hoc Newman–Keuls' test: $p < 0.001$) and RTs were also significantly faster when the tactile stimulus occurred at T8 than at T7 (post hoc Newman–Keuls' test: $p < 0.001$). Moreover, RTs were significantly faster when the tactile stimulus was delivered at T7, T8 and T9 as compared to when the tactile stimulus was delivered at T2, T3, T4, T5 and T6 (post hoc Newman–Keuls' test: $p < 0.001$ in all cases). Analysis also revealed a significant main effect of the SOCIAL CONDITION ($F_{1,27} = 28.90$, $p < 0.001$, $\eta_p^2 = 0.517$). Participants' RTs were significantly faster when participants performed the task in a dyad than alone (post hoc Newman–Keuls' test: $p < 0.001$) (figure 3b). There was no significant effect of the factor PARTNER POSITION.

Those results show that, when participants performed the task in competition with another individual, their RTs were speeded-up. This effect confirmed that participants took into account the

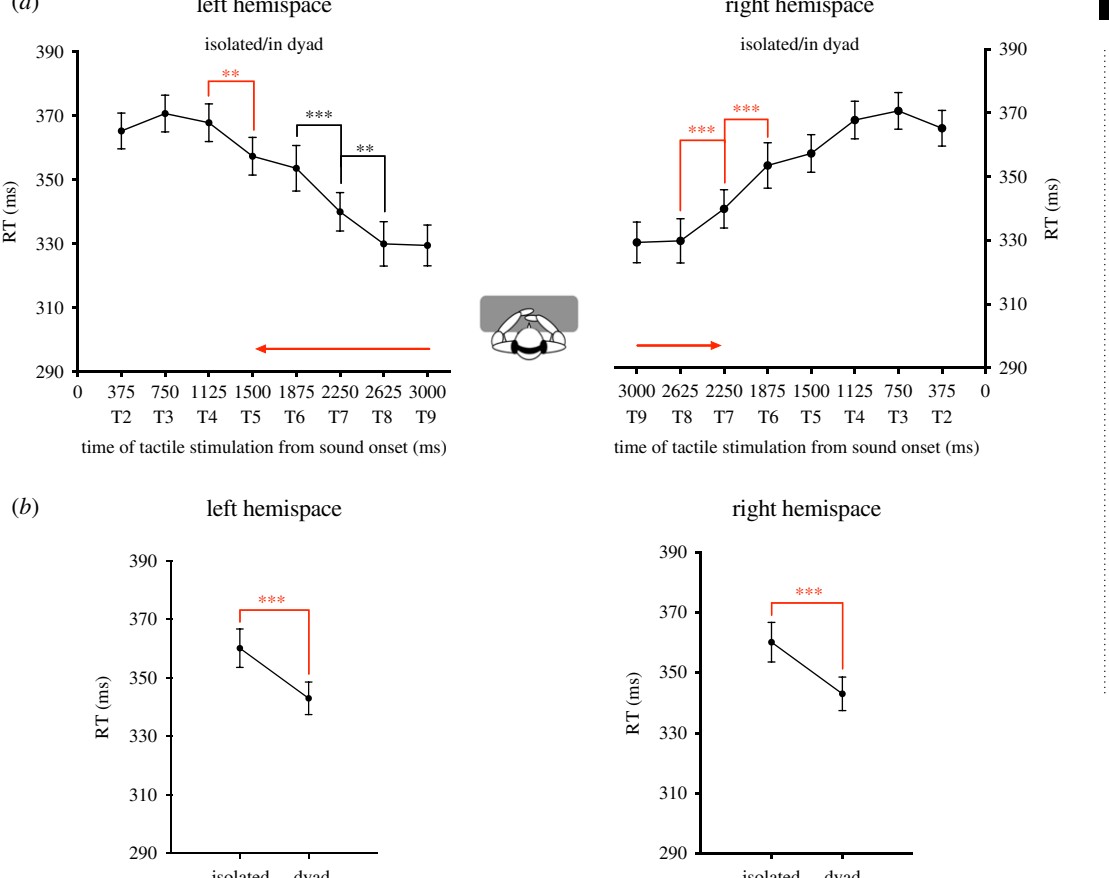

**Figure 3.** Competitive context globally speeds up RTs but does not influence PPS extent. (*a*) This figure reports the mean tactile reaction times (±s.e.m.) as a function of the delay of tactile stimulation delivery from sound onset for the left hemispace (left graph) and for the right hemispace (right graph). There was no effect of the social condition on RTs, thus participants' data in the condition isolated and in dyad are pooled. The significant decreases of RTs are indicated by means of asterisks. The significant decrease of RTs corresponding to the critical delay at which sound starts to boost tactile processing is indicated by means of red asterisks. The red arrows illustrate the extent of PPS boundaries from participants' body. PPS extent is not modified in competitive context. (*b*) This figure reports the mean tactile reaction times (±s.e.m.) for the different social conditions—in isolation and in dyad—in left hemispace (left graph) and for the right hemispace (right graph). In the left and the right hemispace, RTs are significantly faster when participants performed the task in competition with another individual than alone.

competitive instruction. However, lateral PPS extent was not modulated by the presence of a rival. This suggests that PPS extent in competitive context is similar to PPS extent in isolation.

## 3.3. Experiment 3: audience

In this experiment, we tested the impact of the presence of another individual on participants' PPS. Participants performed the audiotactile task either isolated or seated side-by-side with an unknown inactive individual. We conducted separated ANOVA for the left and the right hemispace, with the between subject factor PARTNER POSITION (two levels: Left/Right) and the within subject factors SOCIAL CONDITION (two levels: Isolated/In Dyad), and DELAY (eight levels: T2, T3, T4, T5, T6, T7, T8, T9).

In the left hemispace, analysis indicated a main effect of DELAY ($F_{7,168} = 24.93$, $p < 0.001$, $\eta_p^2 = 0.509$). The other factors and interactions were not significant. Sound in the left hemispace started to boost tactile detection when tactile stimulation is administered at T5 (figure 4). Participants' RTs were significantly faster when the tactile stimulus occurred at T5 compared to when the tactile stimulus occurred at T4 (*post hoc* Newman–Keuls' test: $p < 0.05$). RTs further decreased at the later delays. Participants' RTs were significantly faster when the tactile stimulus occurred at T7 than at T6 (*post hoc* Newman–Keuls' test: $p < 0.01$) and RTs were also significantly faster when the tactile stimulus occurred at T8 than at T7 (*post hoc* Newman–Keuls' test: $p < 0.05$). Moreover, RTs were significantly faster when the tactile stimulus was delivered at T5, T6, T7, T8 and T9 as compared to when the tactile stimulus was

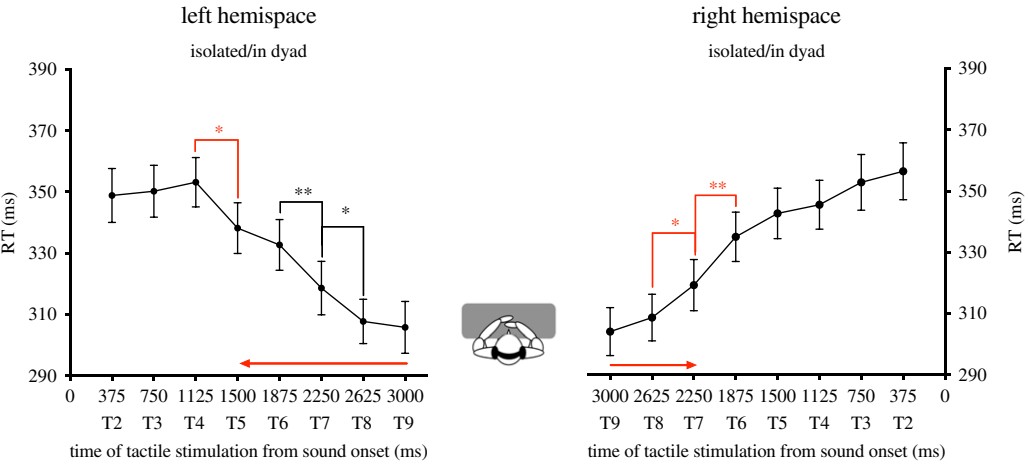

**Figure 4.** Lack of influence of another individual's presence on PPS extent. This figure reports the mean tactile reaction times (±s.e.m.) as a function of the delay of tactile stimulation delivery from sound onset for the left hemispace (left graph) and for the right hemispace (right graph). There was no effect of the social condition on RTs, thus participants' data in the condition isolated and in dyad are pooled. The significant decreases of RTs are indicated by means of asterisks. The significant decrease of RTs corresponding to the critical delay at which sound starts to boost tactile processing is indicated by means of red asterisks. The red arrows illustrate the extent of PPS boundaries from participants' body. PPS extent is not modified by the presence of another individual.

delivered at T2, T3 and T4 (*post hoc* Newman–Keuls' test: $p < 0.05$ in all cases), except for T3 which is only marginally different from T5 (*post hoc* Newman–Keuls' test: $p = 0.067$). There was no significant effect of the factor PARTNER POSITION.

In the right hemispace, analysis indicated a main effect of DELAY ($F_{7,168} = 32.03$, $p < 0.001$, $\eta_p^2 = 0.572$). The other factors and interactions were not significant. Sound in the right hemispace started to boost tactile detection when tactile stimulation is administrated at T7. Participants' RTs were significantly faster when the tactile stimulus occurred at T7 compared to when the tactile stimulus occurred at T6 (*post hoc* Newman–Keuls' test: $p < 0.01$). RTs further decreased at the later delays. Participants' RTs were significantly faster when the tactile stimulus occurred at T8 than at T7 (*post hoc* Newman–Keuls' test: $p < 0.05$). Moreover, RTs were significantly faster when the tactile stimulus was delivered at T7, T8 and T9 as compared to when the tactile stimulus was delivered at T2, T3, T4 and T5 T6 (*post hoc* Newman–Keuls' test: $p < 0.01$ in all cases). There was no significant effect of the factor PARTNER POSITION.

Those results show no effect of the presence an inactive individual on participants' RTs during the audiotactile task. This suggests that participants' lateral PPS extent remained unchanged in an audience context. This also confirms that the observed social effect in the collaborative experiment is not due to the mere presence of another individual, but to the instructions manipulating the social relation between the two individuals.

## 4. Discussion

We investigated whether high-level social manipulation influences the integration of the multisensory information related to a stimulus approaching the body, determining PPS boundaries. We designed experiments to test if the presence of an unknown individual at the left or the right side of the participant modulates PPS extension in the left and right hemispaces. We manipulated group membership between individuals as well as the location of the individuals, while measuring PPS boundaries of participants when they were performing a task in a dyad, in collaborative, competitive or audience contexts. Each experiment was compared to an isolated condition, during which the participant was alone to perform the task.

We first verified that PPS is asymmetrical in the left and right hemispaces, which led us to analyse separately the effect of the social manipulation in each hemispace. When forming a group sharing an intention, individuals in dyads exhibited a boost of audiotactile integration that was further away from the body on the right side in comparison to when performing alone. Surprisingly, this PPS modulation was independent of the position of the partner in space, be it at participant's left or the

right side. Simply affiliating two persons who did not know each other influenced the multisensory coding of the space around them.

We also found evidence for a process of competition in the second experiment, since an impressive global diminution of RT to detect the tactile stimulus was observed. However, the two lateral PPS extents were similar to the ones measured when individuals were in isolation. These results are coherent with the ones in Teneggi et al., who found a PPS modulation in the presence of a cooperative individual but not with a non-cooperative one [27]. Overall, those observations contribute to the literature indicating that affiliative contexts modulate multisensory integration near the body [26–28]. It is also in coherence with the larger literature indicating that social factors modulate the way we apprehend the space around us [39], be it the perception of reachable space [23], affordances [40], interpersonal distances [41,42] or defensive responses to threat in space [43]. In contrast, the situation of audience characterized here by the presence of an inactive individual with whom no relationship was declared, did not introduce any behavioural changes, as already reported for reaching actions [37]. Multisensory integration of stimuli related to the body remained similar to in the situation where individuals were isolated.

The PPS modulation that we observed during collaboration was not dependent on the partner position in space. This finding was unexpected and the potential explanations for the observed pattern merit further investigation. PPS modulations in presence of others have been interpreted as an inclusion of others in our own PPS in affiliative contexts [27,28], or as a mechanism allowing the presence of others near the body during joint actions [26]. Our results point to another direction, as the PPS modulation we found was not related to the partner location. First, this difference may be linked to the social manipulations used in previous studies [27,28], in which the participants' social categorization of the other was based on the quality of social interaction or on moral judgement. In contrast, in the present study we manipulated participants' identification to a group through instructions. Second, previous studies measured the post-effect of social categorization on PPS while participants were in presence of the other individual. We measured the real-time effect of our social manipulation on PPS since both participants performed the task as a group during the measure.

Why would the dynamic coding of PPS in the right hemispace be modified in a collaborative context, whatever the seating side of the confederate? This pattern of results can be linked to theories from the field of social psychology, which propose that in individual members of a group, there is a cognitive transformation from personal to social level identification [44,45]. The modulation of the multisensory coding of PPS observed in the collaborative context might reflect the transition of participants' individual identification towards a group identification shared with the partner. The dynamic coding of PPS is anisotropic in isolated right-handed individuals [30]. By contrast, PPS modulation observed in a collaborative context goes against the natural anisotropy of PPS linked to handedness. As we verified here, right-handed individuals have an asymmetric PPS, right boundaries being closer to the body than left boundaries. The brain and behavioural asymmetry in humans and other vertebrates are thought to give an evolutionary advantage [46]. However, in the collaborative experiment, participants' PPS were more symmetrical. In social psychology, it has been observed that individuals involved in a group sometimes think and act in a way that seems logical at the group level while appearing irrational, even detrimental, at the individual level [47,48]. Similar phenomena could also be present at the level of sensory integration near the body where the seemingly disadvantageous symmetrical remapping of PPS could actually be beneficial for group behaviour.

First, it is now clear that PPS is coded in action terms, with its extent dependent on individuals' action abilities [49–51]. Studies on joint actions show that co-actors represent each other's action in space [14,15], which suggests a new definition of individuals' action space during collaboration. Thus, given that PPS is related to individuals' motor space, we argue that PPS in collaborative context is no longer dependent on individuals' action abilities. Individuals' own motor abilities would not be enough to represent the individual's action space, and it might be that the social context needs to be taken into account to accommodate to the environment. The asymmetric modulation of PPS could be the mark of a complex reorganization of individuals' motor space during collective actions.

Second, our experiments used looming sounds. A looming event is the sign of possible body threats, and can elicit the production of defensive behaviours [52–55]. PPS modulations in our experiments may be linked to defensive/protective behaviours [31]. In isolation, right-handers' PPS is larger in the hemispace of their non-dominant hand, where action implementation is slower and less precise, and thus requires a larger protection margin [30]. In the collaborative experiment, the PPS extension of the right hemispace could be the mark of an extended protection space for the group. Individuals may enlarge their defensive action space in the hemispace of their dominant hand in order to be able to

use it to defend a group member. This proposal is speculative, but it is consistent with a study suggesting that the area of defensive reactions at potential threats could increase in the presence of another individual [43]. The absence of extension in the left hemispace may be linked to a ceiling effect: since the left PPS is already larger for right-handed individuals, it may not require to be further enlarged in social conditions.

Future research is needed to replicate the present findings, and to further understand the links between social interaction and PPS plasticity. It is important to note that in our experiments, sound space is not shared between participants but centred on each individual. Participants listened to sounds looming towards their own body, through headphones. It would be interesting to know if the results are similar in a configuration were the sound space is shared between participants, and in which sounds are looming alternatively towards one participant and towards her partner. Furthermore, the current findings are based on a protocol that rests on tactile detection. The tactile modality has a paramount role in the interaction between individuals and the world around them, in communication and social bonding [56–58]. It is therefore possible that protocols addressing PPS through the measure of the integration of an external stimulus with a tactile stimulation are more prone to evidence a modulation according to the social context. It would be worth replicating the study with other sensory modalities to account for a specific role of touch in the present study.

Group membership seems to be sufficient to modulate the sensory coding of the space around interacting individuals in a way that would not have been hypothesized based on the studies conducted on isolated individuals. It has been proposed that generalizing claims about cognition within the individual to social cognition cannot explain the relation between mind and society [48,59,60]. There is yet much more to be done in understanding the underpinnings of the relationship between social context and the coding of multisensory information related to the bodies in space. Nevertheless, the present study goes along with this idea by suggesting that the mechanisms of multisensory integration within the individual cannot be generalized to the mechanisms of multisensory integration in social context. One possible explanation is that the demands of social interaction shape low-level physiological mechanisms, such as the way different sensory information is integrated to make sense of the world around us.

Ethics. All participants provided a written informed consent prior to the experiment, which was approved by the local ethics committee: the Institutional Review Board of the French National Institute of Health and Medical Research (INSERM, IRB00003888).

Data accessibility. Data collected for this study can be found in the Dryad Digital Repository at: https://doi.org/10.5061/dryad.76980vs [61].

Authors' contributions. L.H., M.T. and I.V.-D. designed the experiments reported in the paper. L.H. collected the data; L.H. and M.T. performed the analyses. L.H., M.T. and I.V.-D. wrote the article.

Competing interests. The authors declare no competing interests.

Funding. This work was supported by the funding of Sorbonne Universités Investissements d'avenir, Emergence; and by the programme Bettencourt of the FdV doctoral school (Ecole Doctorale Frontières du Vivant (FdV)—Programme Bettencourt).

Acknowledgements. We are grateful to Emmanuel Fléty and Arnaud Recher for their help with the apparatus for tactile stimulation. We thank Olivier Warusfel for his help on the elaboration of spatialized auditory stimuli through binaural rendering. We thank Philippe Nivaggioli for his help during set-up installation, and Anna Skrzatek for her help with data collection. We thank Alice Castan for proofreading this manuscript for British English spelling.

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

**13**