## [Reviewer comments · Royal Society Open Science]

Review History

RSOS-181878.R0 (Original submission)

Review form: Reviewer 1 (Flavia Cardini)

Is the manuscript scientifically sound in its present form?

Yes

Are the interpretations and conclusions justified by the results?

Yes

Is the language acceptable?

Yes

Is it clear how to access all supporting data?

Yes

Do you have any ethical concerns with this paper?

No

Have you any concerns about statistical analyses in this paper?

No

Recommendation?

Major revision is needed (please make suggestions in comments)

Comments to the Author(s)

General comments

In their manuscript Hobeika and colleagues investigated the effect of social interaction in the immediate reshaping of the PPS. They took advantage of a widely used paradigm, slightly manipulated to answer their research question.

The results showed that only during collaboration with a partner the boundaries of the PPS expand in the right hemisphere.

The idea is very interesting and the study very well designed.

Incredibly clear and detailed methods section. It is the first manuscript I read in my life where I do not need to read the methods section twice to grasp the whole methodology. I really thank the authors for their clarity.

I have some comments that I am listing below, and that I would like the authors to address before recommending the manuscript for publication.

Major

It is not clear to me why the analyses use the hemisphere variable (left vs right) instead of the side where the confederate was sitting. In line with Teneggi's study, I would expect the PPS to expand after cooperative activity towards the side of the confederate. Therefore I would expect an initial analysis to investigate this hypothesis. Could authors please justify the choice of analysing the hemispaces instead of the confederate location? I do understand that given that lateral PPS is asymmetric for right-handers, it makes sense to analyse separately each hemisphere. But why not considering whether in that hemisphere there was a confederate sitting or not?

I could not find clear hypotheses about the modulation of the PPS in each hemisphere based on the confederate presence. Without some hypotheses, the analyses presented by the authors sound quite post-hoc.

Figure 2. In the RH participants performed differently when in dyad as compared to when in isolation. In the "dyad" graph authors report that "The significant decrease of RTs corresponding to the critical delay at which sound starts to boost tactile processing is indicated by means of red asterisks." How did authors decide that the boundaries of PPS is between T4 and T3, and not for example between T8 and T6, where another significant difference in RTs was found?

I did not find really exhaustive the explanation of the PPS expansion in the right hemisphere during collaborative activity. Could authors please try to speculate more about why the right and not the left hemisphere is reshaped during a collaborative interaction?

And also, why not the hemisphere where the confederate was sitting?

Minor

Page 8-9, lines 212-213. Font is different.

Font in figures is not consistent.

Page 8, line 249. "Expect" should be "except"

Review form: Reviewer 2

Is the manuscript scientifically sound in its present form?

Yes

Are the interpretations and conclusions justified by the results?

No

Is the language acceptable?

Yes

Is it clear how to access all supporting data?

Yes

Do you have any ethical concerns with this paper?

No

Have you any concerns about statistical analyses in this paper?

Yes

Recommendation?

Major revision is needed (please make suggestions in comments)

Comments to the Author(s)

This paper investigates an audiotactile interaction task for stimuli in or out peripersonal space (PPS) presented in a social context. Participants had to detect as fast as possible a tactile stimulus while task-irrelevant looming sounds were presented, when they were paired as collaborative or competitive dyads, or in the presence of an inactive individual. Results suggest that PPS extended when participants collaborated with a partner, obviously to adapt individual motor control to an interaction with a social context. This underlines the effect of the need to interact with others on low-level spatial perception.

The papers presents a well conducted study with interesting outcome. There is several points that need to be considered before publication. First, the originality of the study is mitigated by the lack of quotation of publications that are relevant with respect to the research topic. In particular, one paper (and a chapter) has addressed the same issue and needs to be quoted and discussed. Coello, Y., Quesque, F., Gigliotti, M-F., Ott, L., Bruyelle, J-L. (2018). Idiosyncratic representation of peripersonal space depends on the success of one's own motor actions, but also the successful actions of others! PLOS ONE, <https://doi.org/10.1371/journal.pone.0196874>.

Coello, Y. (2018). Action Spaces Representation in Social Contexts. In Shigemasa, K., Kuwano, S., Sato, T., Matsuzawa, T. (Eds.), *Diversity in Harmony - Insights from Psychology: Proceedings of the 31st International Congress of Psychology*. Wiley. <https://doi.org/10.1002/9781119362081.fmatter>.

Furthermore, concerning social and affective processes involved in PPS, several publications by the group of Iachini have been evaded:

Cartaud, A., Ruggiero, G., Ott, L., Iachini, T., Coello, Y. (2018). Physiological response to facial expressions in peripersonal space determines interpersonal distance in a social interaction context. *Frontiers Psychology*, 9, 657. doi: 10.3389/fpsyg.2018.00657

Quesque, F., Ruggiero, G., Mouta, S., Santos, J., Iachini, T., Coello, Y. (2017). Keeping you at arm's length: Modifying peripersonal space influences interpersonal distance. *Psychological Research*, 81, (4), 709–720.

Ruggiero, G., Frassinetti, F. Coello, Y., Rapuano, C., Schiano di Cola, A., Iachini, T. (2016). The effect of facial expressions on peripersonal and interpersonal spaces. *Psychological Research*. doi: 10.1007/s00426-016-0806-x.

Iachini, T., Coello, Y., Frassinetti, F., Senese, V.P., Galante, F., Ruggiero, G. (2016). Peripersonal and interpersonal space in virtual and real environments: effects of gender and age. *Journal of*

Environmental Psychology, 45, 154-164.

Iachini, T., Coello, Y., Frassinetti, F., Ruggiero, G. (2014). Body space in social interactions: A comparison of reaching and comfort distance in immersive virtual reality. *PlosOne*, 9(11), e111511.

Coello, Y., Bourgeois, J., Iachini, T. (2012). Embodied perception of reachable space: How do we manage threatening objects? *Cognitive Processing*, 1, 131-135.

Concerning PPS and Morality:

Iachini, T., Pagliaro, S., Ruggiero, G. (2015). Near or far? It depends on my impression: moral information and spatial behavior in virtual interactions. *Acta Psychol (Amst)*. 2015 Oct;161:131-6.

The second point is that the data analysis should consider all variables. The statistical analysis should be performed including the side effect and I recommend performing a 3 way Anova including Condition x Side x SOA.

The third point is that the interpretation of the data are in some aspects overinterpreting the results. It is indeed no fully obvious that in Exp 1 "... results show that PPS right boundaries were farther away when participants were performing the task together with another individual than when they were isolated. "

- The comparisons are made only between SOAs, which cannot guaranty against artefacts in the context of multiple comparisons. It is thus important to show that RT also changes across conditions for similar SOA.

- The fact that there is no effect of the location of the confederate (right or left) on the extent of right PPS weakened the interpretation that is proposed. What is observed is that, in both the individual and social conditions, there is a decreasing slope of RT depending on the distance, but with non-equivalent local effects. The slope is steeper in the social condition (which could be tested), claiming for a boost of RT in the social context. However, the lack of effect on the left side makes the interpretation of the results difficult. Furthermore, the boost effect is observed both sides in the competition condition. Accordingly, claiming that "PPS extent is sensitive to a high-level social instruction of collaboration" seems overstepping, considering that this is actual only on the right side (Exp1). Although the conclusions should be tempered, the fact that the effect was not observed on the left side could be related to the fact that the response was always delivered with the right hand. A control experiment with the response provided by the left hand could be valuable to better understand the articulation of the effects reported. It is also not clear why a side by side condition will produce an extension of PPS. Extension of PPS was obtained in a social context when two facing confederates performed a collaborative task (Teneggi et al. 2013; Coello et al., 2018). Perhaps, this could be argued.

Minor points

Method

- What is the role of the visual target? How long was it displayed?

- Concerning the conditions, there is 2 contexts (isolated, social), 2 directions (right, left) and 11 time delays. I thus wouldn't say that there is 22 conditions (but 44), with 10 trials in each. Also, the authors mention 22 conditions with 10 target stimuli in each, it is thus not clear why there is 440 trials.

- Concerning the seating condition, (left/right) it must be indicated if this condition was within/between subjects.

Figure 1 shows an exact location for T1 to T9 according to the participant. How were these location determined (time to contact?), were they perceived similarly by all participants (it is indicated that "The absolute distance estimation of sound sources is usually a difficult task for non-familiar stimuli in absence of reference, in which participants are not accurate). It would be interested to have an information concerning at which distance multisensory integration occurs for each participant.

- Page 7. "RTs were significantly faster when the tactile stimulus occurred at T4 compared to

when the tactile stimulus occurred at T5". This is counterintuitive and seems the opposite to what is reported Figure 2.

Decision letter (RSOS-181878.R0)

10-May-2019

Dear Dr Hobeika,

The editors assigned to your paper ("Social coding of the multisensory space around us") have now received comments from reviewers. We would like you to revise your paper in accordance with the referee and Associate Editor suggestions which can be found below (not including confidential reports to the Editor). Please note this decision does not guarantee eventual acceptance.

Please submit a copy of your revised paper before 02-Jun-2019. Please note that the revision deadline will expire at 00.00am on this date. If we do not hear from you within this time then it will be assumed that the paper has been withdrawn. In exceptional circumstances, extensions may be possible if agreed with the Editorial Office in advance. We do not allow multiple rounds of revision so we urge you to make every effort to fully address all of the comments at this stage. If deemed necessary by the Editors, your manuscript will be sent back to one or more of the original reviewers for assessment. If the original reviewers are not available, we may invite new reviewers.

- Data accessibility

It is a condition of publication that all supporting data are made available either as supplementary information or preferably in a suitable permanent repository. The data accessibility section should state where the article's supporting data can be accessed. This section should also include details, where possible of where to access other relevant research materials

such as statistical tools, protocols, software etc can be accessed. If the data have been deposited in an external repository this section should list the database, accession number and link to the DOI for all data from the article that have been made publicly available. Data sets that have been deposited in an external repository and have a DOI should also be appropriately cited in the manuscript and included in the reference list.

If you wish to submit your supporting data or code to Dryad (<http://datadryad.org/>), or modify your current submission to dryad, please use the following link:
<http://datadryad.org/submit?journalID=RSOS&manu=RSOS-181878>

- **Competing interests**

- **Authors' contributions**

- **Acknowledgements**

- **Funding statement**

Kind regards,

Andrew Dunn

on behalf of Dr Stephen Jackson (Associate Editor) and Essi Viding (Subject Editor)

Associate Editor's comments (Dr Stephen Jackson):

Two reviewers have provided comments on your paper, and we would like you to revise the paper to respond to their commentary. Please also ensure you provide a full point-by-point response outlining the changes made or any rebuttals you feel are appropriate.

Comments to Author:

Reviewers' Comments to Author:

Reviewer: 1

Comments to the Author(s)

General comments

In their manuscript Hobeika and colleagues investigated the effect of social interaction in the immediate reshaping of the PPS. They took advantage of a widely used paradigm, slightly manipulated to answer their research question.

The results showed that only during collaboration with a partner the boundaries of the PPS expand in the right hemisphere.

The idea is very interesting and the study very well designed.

Incredibly clear and detailed methods section. It is the first manuscript I read in my life where I do not need to read the methods section twice to grasp the whole methodology. I really thank the authors for their clarity.

I have some comments that I am listing below, and that I would like the authors to address before recommending the manuscript for publication.

Major

It is not clear to me why the analyses use the hemisphere variable (left vs right) instead of the side where the confederate was sitting. In line with Teneggi's study, I would expect the PPS to expand after cooperative activity towards the side of the confederate. Therefore I would expect an initial analysis to investigate this hypothesis. Could authors please justify the choice of analysing the hemispaces instead of the confederate location? I do understand that given that lateral PPS is asymmetric for right-handers, it makes sense to analyse separately each hemisphere. But why not considering whether in that hemisphere there was a confederate sitting or not?

I could not find clear hypotheses about the modulation of the PPS in each hemisphere based on the confederate presence. Without some hypotheses, the analyses presented by the authors sound quite post-hoc.

Figure 2. In the RH participants performed differently when in dyad as compared to when in isolation. In the "dyad" graph authors report that "The significant decrease of RTs corresponding to the critical delay at which sound starts to boost tactile processing is indicated by means of red asterisks." How did authors decide that the boundaries of PPS is between T4 and T3, and not for example between T8 and T6, where another significant difference in RTs was found?

I did not find really exhaustive the explanation of the PPS expansion in the right hemisphere during collaborative activity. Could authors please try to speculate more about why the right and not the left hemisphere is reshaped during a collaborative interaction?

And also, why not the hemisphere where the confederate was sitting?

Minor

Page 8-9, lines 212-213. Font is different.

Font in figures is not consistent.

Page 8, line 249. "Expect" should be "except"

Reviewer: 2

Comments to the Author(s)

This paper investigates an audiotactile interaction task for stimuli in or out peripersonal space (PPS) presented in a social context. Participants had to detect as fast as possible a tactile stimulus while task-irrelevant looming sounds were presented, when they were paired as collaborative or competitive dyads, or in the presence of an inactive individual. Results suggest that PPS extended when participants collaborated with a partner, obviously to adapt individual motor control to an interaction with a social context. This underlines the effect of the need to interact with others on low-level spatial perception.

The papers presents a well conducted study with interesting outcome. There is several points that need to be considered before publication. First, the originality of the study is mitigated by the lack of quotation of publications that are relevant with respect to the research topic. In particular, one paper (and a chapter) has addressed the same issue and needs to be quoted and discussed. Coello, Y., Quesque, F., Gigliotti, M-F., Ott, L., Bruyelle, J-L. (2018). Idiosyncratic representation of peripersonal space depends on the success of one's own motor actions, but also the successful actions of others! PLOS ONE, <https://doi.org/10.1371/journal.pone.0196874>. Coello, Y. (2018). Action Spaces Representation in Social Contexts. In Shigemasa, K., Kuwano, S., Sato, T., Matsuzawa, T. (Eds.), *Diversity in Harmony – Insights from Psychology: Proceedings of the 31st International Congress of Psychology*. Wiley. <https://doi.org/10.1002/9781119362081.fmatter>.

Furthermore, concerning social and affective processes involved in PPS, several publications by the group of Iachini have been evaded:

- Cartaud, A., Ruggiero, G., Ott, L., Iachini, T., Coello, Y. (2018). Physiological response to facial expressions in peripersonal space determines interpersonal distance in a social interaction context. *Frontiers Psychology*, 9, 657. doi: 10.3389/fpsyg.2018.00657
- Quesque, F, Ruggiero, G., Mouta, S., Santos, J., Iachini, T., Coello, Y. (2017). Keeping you at arm's length: Modifying peripersonal space influences interpersonal distance. *Psychological Research*, 81, (4), 709–720.
- Ruggiero, G., Frassinetti, F. Coello, Y., Rapuano, C., Schiano di Cola, A., Iachini, T. (2016). The effect of facial expressions on peripersonal and interpersonal spaces. *Psychological Research*. doi: 10.1007/s00426-016-0806-x.
- Iachini, T., Coello, Y., Frassinetti, F., Senese, V.P., Galante, F., Ruggiero, G. (2016). Peripersonal and interpersonal space in virtual and real environments: effects of gender and age. *Journal of Environmental Psychology*, 45, 154-164.
- Iachini, T., Coello, Y., Frassinetti, F., Ruggiero, G. (2014). Body space in social interactions: A comparison of reaching and comfort distance in immersive virtual reality. *PlosOne*, 9(11), e111511.
- Coello, Y., Bourgeois, J., Iachini, T. (2012). Embodied perception of reachable space: How do we manage threatening objects ? *Cognitive Processing*, 1, 131-135.
- Concerning PPS and Morality:
- Iachini, T., Pagliaro, S., Ruggiero, G. (2015). Near or far? It depends on my impression: moral information and spatial behavior in virtual interactions. *Acta Psychol (Amst)*. 2015 Oct;161:131-6.

The second point is that the data analysis should consider all variables. The statistical analysis should be performed including the side effect and I recommend performing a 3 way Anova including Condition x Side x SOA.

The third point is that the interpretation of the data are in some aspects overinterpreting the results. It is indeed no fully obvious that in Exp 1 "... results show that PPS right boundaries

were farther away when participants were performing the task together with another individual than when they were isolated. “

- The comparisons are made only between SOAs, which cannot guaranty against artefacts in the context of multiple comparisons. It is thus important to show that RT also changes across conditions for similar SOA.

- The fact that there is no effect of the location of the confederate (right or left) on the extent of right PPS weakened the interpretation that is proposed. What is observed is that, in both the individual and social conditions, there is a decreasing slope of RT depending on the distance, but with non-equivalent local effects. The slope is steeper in the social condition (which could be tested), claiming for a boost of RT in the social context. However, the lack of effect on the left side makes the interpretation of the results difficult. Furthermore, the boost effect is observed both sides in the competition condition. Accordingly, claiming that “PPS extent is sensitive to a high-level social instruction of collaboration” seems overstepping, considering that this is actual only on the right side (Exp1). Although the conclusions should be tempered, the fact that the effect was not observed on the left side could be related to the fact that the response was always delivered with the right hand. A control experiment with the response provided by the left hand could be valuable to better understand the articulation of the effects reported. It is also not clear why a side by side condition will produce an extension of PPS. Extension of PPS was obtained in a social context when two facing confederates performed a collaborative task (Teneggi et al. 2013; Coello et al., 2018). Perhaps, this could be argued.

Minor points

Method

- What is the role of the visual target ? How long was it displayed ?

- Concerning the conditions, there is 2 contexts (isolated, social), 2 directions (right, left) and 11 time delays. I thus wouldn't say that there is 22 conditions (but 44), with 10 trials in each. Also, the authors mention 22 conditions with 10 target stimuli in each, it is thus not clear why there is 440 trials.

- Concerning the seating condition, (left/right) it must be indicated if this condition was within/between subjects.

Figure 1 shows an exact location for T1 to T9 according to the participant. How were these location determined (time to contact ?), were they perceived similarly by all participants (it is indicated that “ The absolute distance estimation of sound sources is usually a difficult task for non-familiar stimuli in absence of reference, in which participants are not accurate). It would be interested to have an information concerning at which distance multisensory integration occurs for each participant.

- Page 7. “RTs were significantly faster when the tactile stimulus occurred at T4 compared to when the tactile stimulus occurred at T5”. This is counterintuitive and seems the opposite to what is reported Figure 2.

Author's Response to Decision Letter for (RSOS-181878.R0)

See Appendix A.

Decision letter (RSOS-181878.R1)

02-Jul-2019

Dear Dr Hobeika,

I am pleased to inform you that your manuscript entitled "Social coding of the multisensory space around us" is now accepted for publication in Royal Society Open Science.

Kind regards,

on behalf of Dr Stephen Jackson (Associate Editor) and Essi Viding (Subject Editor)
openscience@royalsociety.org

Appendix A

We would like to express our gratitude to the reviewers for their comments. Thanks to them, we have understood that one of the main experimental condition of the study, i.e. the effect of the partner position in space, was confusing. *We have modified the manuscript to clarify the theoretical importance of this factor.* We also modified the name of this factor for the statistical analysis throughout the manuscript. It was previously called “ SEATING”, and is now called “PARTNER POSITION”.

Reviewer: 1

Comments to the Author(s)

General comments

In their manuscript Hobeika and colleagues investigated the effect of social interaction in the immediate reshaping of the PPS. They took advantage of a widely used paradigm, slightly manipulated to answer their research question.

The results showed that only during collaboration with a partner the boundaries of the PPS expand in the right hemisphere.

The idea is very interesting and the study very well designed.

Incredibly clear and detailed methods section. It is the first manuscript I read in my life where I do not need to read the methods section twice to grasp the whole methodology. I really thank the authors for their clarity.

I have some comments that I am listing below, and that I would like the authors to address before recommending the manuscript for publication.

Major

1) It is not clear to me why the analyses use the hemisphere variable (left vs right) instead of the side where the confederate was sitting. In line with Teneggi's study, I would expect the PPS to expand after cooperative activity towards the side of the confederate. Therefore I would expect an initial analysis to investigate this hypothesis. Could authors please justify the choice of analysing the hemispaces instead of the confederate location? I do understand that given that lateral PPS is asymmetric for right-handers, it makes sense to analyse separately each hemisphere. But why not considering whether in that hemisphere there was a confederate sitting or not? I could not find clear hypotheses about the modulation of the PPS in each hemisphere based on the confederate presence. Without some hypotheses, the analyses presented by the authors sound quite post-hoc.

In our study, we wanted to examine if the presence of a cooperative, competitive or inactive partner modulates lateral PPS. We expected that we would observe a modulation of PPS boundaries in social contexts, and this modulation would be dependent of the valence of the interaction. In line with Teneggi et al (2013) we predicted that PPS would expand in presence of a cooperative partner, and contract in presence of a competitive one. As reviewer 1, we expected this expansion to be observed only in the direction of the partner. We added this hypothesis on the last but one paragraph of the introduction (lines 97-112)

In order to test this hypothesis, we included 4 experimental factors in the experiments. There were three within subject factors: HEMISPHERE (2 levels: Left/Right), which corresponds to the location of the sound source in space, SOCIAL CONDITION (2 levels: Isolated/In Dyad), which refers to the fact that participants performed the task alone or in presence of a partner; and DELAY (8 levels: T2, T3, T4, T5, T6, T7, T8, T9), which refers to the delay between the sound onset and the time of tactile delivery, and is related to the distance of the sound source in space. There was one between subject factor: the PARTNER POSITION (2 levels: Left/Right) which corresponds to the fact that the partner could be seated at the left or the right side of the participant in the dyad condition.

In line with previous work, we expected lateral PPS to be asymmetrical for right-handers participants (Hobeika et al. 2018), with a larger PPS in the left hemispace than in the right hemispace. This asymmetry was replicated in the current study. Thus, as audiotactile integration is different in left and right hemispaces when participants are alone, we tested its social modulation separately in the two hemispaces. This choice allowed to increase the statistical power of the analysis, considering the large number of conditions in the experiments. We performed an ANOVA on the data obtained in each hemispace, including the within subjects factors SOCIAL CONDITION and DELAY, and the between subjects factor PARTNER POSITION.

The analysis tested if potential PPS social modulations were in direction of the partner, by including the factor PARTNER POSITION. There was no main effect of the factor PARTNER POSITION, and no interaction of this factor with the SOCIAL CONDITION. Therefore, we obtained the surprising result that PPS was modulated in collaborative contexts only in the right hemispace, and independently of the partner position.

2) Figure 2. In the RH participants performed differently when in dyad as compared to when in isolation. In the “dyad” graph authors report that “The significant decrease of RTs corresponding to the critical delay at which sound starts to boost tactile processing is indicated by means of red asterisks.” How did authors decide that the boundaries of PPS is between T4 and T3, and not for example between T8 and T6, where another significant difference in RTs was found?

PPS is the space close to the body coded by multisensory neurons. Multisensory integration leads to a faster detection of stimuli. Thus, multisensory events occurring in the PPS are detected faster than outside the PPS.

To determine the boundary of PPS in space, we take the furthest distance from participants’ body at which the presence of a sound boosted tactile detection. Reaction times further decrease after the first boost, as the sound comes closer within PPS: the distance between the auditory and the tactile stimulus continues to influence tactile reaction time. The closer the two sensory stimuli, the stronger the multisensory boost of the detection times is.

This explanation is in the manuscript in the last paragraph of the introduction, lines 116-121: *“As such, participants perceived the sound source at different distances from their body when they processed the tactile stimulus (from the farthest distance at T1 to the closest distance at T9). As the delay increased, the looming sound was perceived as closer. It is now well established that a sound speeds up tactile detection when located near, but not far from, the body, i.e. within PPS [32,33]. Consequently, we sought for the distance from participants’ body, at which sound starts to boost tactile RTs as a proxy of PPS boundaries.”*

3) I did not find really exhaustive the explanation of the PPS expansion in the right hemispace during collaborative activity. Could authors please try to speculate more about why the right and not the left hemispace is reshaped during a collaborative interaction? And also, why not the hemispace where the confederate was sitting?

We have speculated more about the possible interpretation of the asymmetry in the social modulation of PPS in the discussion, lines 419-430:

«Second, our experiments used looming sounds. A looming event is the sign of possible body threats, and can elicit the production of defensive behaviours [52–54]. PPS modulations in our experiments

may be linked to defensive/protective behaviours [31]. In isolation, right-handers PPS is larger in the hemispace of their non-dominant hand, where actions implementation is slower and less precise, and thus requires a larger protection margin [30]. In the collaborative experiment, the PPS extension of the right hemispace could be the mark of an extended protection space for the group. Individuals may enlarge their defensive action space in the hemispace of their dominant hand in order to be able to use it to defend a group member. This proposal is speculative, but it is consistent with a study suggesting that the area of defensive reactions at potential threats could increase in the presence of another individual [43]. The absence of extension in the left hemispace may be linked to a ceiling effect: since the left PPS is already larger for right-handed individuals, it may not require to be further enlarged in social conditions. »

Minor

Page 8-9, lines 212-213. Font is different.

Font in figures is not consistent.

Page 8, line 249. "Expect" should be "except"

Those corrections have been made in the new version of the manuscript

Reviewer: 2

Comments to the Author(s)

This paper investigates an audiotactile interaction task for stimuli in or out peripersonal space (PPS) presented in a social context. Participants had to detect as fast as possible a tactile stimulus while task-irrelevant looming sounds were presented, when they were paired as collaborative or competitive dyads, or in the presence of an inactive individual. Results suggest that PPS extended when participants collaborated with a partner, obviously to adapt individual motor control to an interaction with a social context. This underlines the effect of the need to interact with others on low-level spatial perception.

1) The papers presents a well conducted study with interesting outcome. There is several points that need to be considered before publication. First, the originality of the study is mitigated by the lack of quotation of publications that are relevant with respect to the research topic. In particular, one paper (and a chapter) has addressed the same issue and needs to be quoted and discussed.

[1] Coello, Y., Quesque, F., Gigliotti, M-F., Ott, L., Bruyelle, J-L. (2018). Idiosyncratic representation of peripersonal space depends on the success of one's own motor actions, but also the successful actions of others! PLOS ONE, <https://doi.org/10.1371/journal.pone.0196874>.

[2] Coello, Y. (2018). Action Spaces Representation in Social Contexts. In Shigemasu, K., Kuwano, S., Sato, T., Matsuzawa, T. (Eds.), Diversity in Harmony – Insights from Psychology: Proceedings of the 31st International Congress of Psychology. Wiley. <https://doi.org/10.1002/9781119362081.fmatter>.

Furthermore, concerning social and affective processes involved in PPS, several publications by the group of Iachini have been evaded:

[3] Cartaud, A., Ruggiero, G., Ott, L., Iachini, T., Coello, Y. (2018). Physiological response to facial expressions in peripersonal space determines interpersonal distance in a social interaction context. *Frontiers Psychology*, 9, 657. doi: 10.3389/fpsyg.2018.00657

[4] Quesque, F., Ruggiero, G., Mouta, S., Santos, J., Iachini, T., Coello, Y. (2017). Keeping you at arm's length: Modifying peripersonal space influences interpersonal distance. *Psychological Research*, 81, (4), 709–720.

[5] Ruggiero, G., Frassinetti, F., Coello, Y., Rapuano, C., Schiano di Cola, A., Iachini, T. (2016). The effect of facial expressions on peripersonal and interpersonal spaces. *Psychological Research*. doi: 10.1007/s00426-016-0806-x.

[6] Iachini, T., Coello, Y., Frassinetti, F., Senese, V.P., Galante, F., Ruggiero, G. (2016). Peripersonal and interpersonal space in virtual and real environments: effects of gender and age. *Journal of Environmental Psychology*, 45, 154-164.

[7] Iachini, T., Coello, Y., Frassinetti, F., Ruggiero, G. (2014). Body space in social interactions: A comparison of reaching and comfort distance in immersive virtual reality. *PlosOne*, 9(11), e111511.

[8] Coello, Y., Bourgeois, J., Iachini, T. (2012). Embodied perception of reachable space: How do we manage threatening objects? *Cognitive Processing*, 1, 131-135.

Concerning PPS and Morality:

[9] Iachini, T., Pagliaro, S., Ruggiero, G. (2015). Near or far? It depends on my impression: moral information and spatial behavior in virtual interactions. *Acta Psychol (Amst)*. 2015 Oct;161:131-6.

Thank you for this remark. We are aware that there are several published studies on the management of space in social contexts. However, just few of those studies specifically concentrate on multisensory integration related to social context, as Heed et al. 2010, Tennegi et al. 2013 and Pellencin et al 2017. We focus the manuscript's introduction on the literature about the link between multisensory integration and the space close to the body to avoid misconceptions, and we have clarified this in the manuscript.

Peripersonal space, which is coded by multisensory neurons, is usually depicted as the space around the body coded in motor terms for the purpose of defensive actions, or for the implementation of reaching actions. Multiple methodologies have been developed to study peripersonal space based on those characteristics. They can be divided in three main categories: multisensory integration based methods, reaching space based methods and defensive function based methods. However, there is no consensus on whether those methods are related to the same cognitive phenomenon, and whether there is a unique representation of the peripersonal space. De Vignemont and Iannetti (2014) argue for the possibility of two different peripersonal spaces, distinguishing for defensive actions and reaching actions. Cléry and colleagues (2014) also support the idea that there could be two different neural systems to support safety actions and reaching actions.

Lastly, peripersonal space is usually discussed in regards to the large literature of social psychology on personal space, i.e. the space near the body in which others cannot intrude without causing discomfort. Personal space is measured with high-level cognitive tasks, in which participants evaluate the distance between them and others with which they feel comfortable to interact, for example to have a conversation. Again, there is no consensus on the fact that personal space and peripersonal space rely on

the same mechanisms. As an example, Patané et al (2017) found distinct modulations of personal space and reachability space after a collaborative task.

The articles suggested by the reviewer are studies on reaching space and the social personal space. We do not want to emphasize on this literature in the introduction to avoid confusion between the concepts. However, we agree with the reviewer that it is important to refer to this literature. In this new version of the manuscript, we refer to the literature on social influence on space processing in the introduction and in the discussion, where we have added and briefly discussed several of the suggested references.

2) The second point is that the data analysis should consider all variables. The statistical analysis should be performed including the side effect and I recommend performing a 3 way Anova including Condition x Side x SOA.

Thanks a lot for pointing out that the analysis did not emphasize enough all the factors taken into account. The first reviewer also made this remark (see point 1). The presentation of our manuscript was indeed not clear enough to understand that the side effect was included in the performed analyses. This has been corrected.

3) The third point is that the interpretation of the data are in some aspects overinterpreting the results. It is indeed no fully obvious that in Exp 1 "... results show that PPS right boundaries were farther away when participants were performing the task together with another individual than when they were isolated. "

- The comparisons are made only between SOAs, which cannot guaranty against artefacts in the context of multiple comparisons. It is thus important to show that RTs also changes across conditions for similar SOA.

In the collaborative experiment, the anova revealed a significant interaction between the Social Condition (isolated/alone) and the Delay (8 levels) in the right hemisphere. To understand this interaction, we performed Newman-Keuls' post-hoc tests. Newman Keuls' post-hoc tests correct for multiple comparisons, thus diminish the probability of observing false positive.

We hypothesized that this significant interaction could be due to a difference in PPS extent between the isolated and dyad conditions. Thus, we determined PPS extent on both social conditions by looking at speed up effects within each condition (based on previous studies analysis like Canzoneri et al 2012). The analysis revealed a difference in PPS extent. Then, we looked at differences in RTs between the social conditions, for each delay separately. We found no significant differences of RTs across social conditions.

We agree with the reviewer that significant differences of RTs for similar delays across social conditions would be an important argument to support our conclusion. However, in the absence of significant differences of RTs across social conditions, the comparison of RTs within each social condition is the best way to explain the significant interaction between the social conditions and the delays.

- The fact that there is no effect of the location of the confederate (right or left) on the extent of right PPS weakened the interpretation that is proposed. What is observed is that, in both the individual and social conditions, there is a decreasing slope of RT depending on the distance, but with non-equivalent local effects. The slope is steeper in the social condition (which could be tested), claiming for a boost of RT in the social context. However, the lack of effect on the left side makes the interpretation of the results difficult. Furthermore, the boost effect is observed both sides in the competition condition. Accordingly, claiming that "PPS extent is sensitive to a high-level social instruction of collaboration" seems overstepping, considering that this is actual only on the right side (Exp1). Although the conclusions should be tempered, the fact that the effect was not observed on the left side could be related to the fact that the response was always delivered with

the right hand. A control experiment with the response provided by the left hand could be valuable to better understand the articulation of the effects reported. It is also not clear why a side by side condition will produce an extension of PPS. Extension of PPS was obtained in a social context when two facing confederates performed a collaborative task (Teneggi et al. 2013; Coello et al., 2018). Perhaps, this could be argued.

We agree that the present results are puzzling and can be seen as controversial at first. However, taking an approach considering the asymmetry of PPS for right-handers, and the potential functional role of PPS, we can propose a congruent explanation to these findings. Here are some answers to the duly justified comments of the reviewer:

-> The fact that there is no effect of the location of the confederate (right or left) on the extent of right PPS weakened the interpretation that is proposed.

The fact that the PPS modulation is not dependent of the partner position is clearly surprising, but to our opinion it does not weaken the results. On the contrary, we think that this result can lead to new interpretations of results of PPS in social contexts, and to new theoretical approaches on the management of space in social contexts. For example, Teneggi et al. (2012) argue that in a cooperative context, PPS extend to include the partner. This hypothesis does not stand if the extension is not dependent of the partner position.

Experiments on social modulations of PPS always place two individuals facing each others, and measure PPS extent in one direction only: in the frontal space where the confederate is located. Thus, it is not possible to conclude if PPS modulations are in the direction of the confederate, or changes in the front whatever the partner position. Our study is the first to tackle this question. We have better explained this in the last but one paragraph of the introduction, lines 97-112.

-> What is observed is that, in both the individual and social conditions, there is a decreasing slope of RT depending on the distance, but with non-equivalent local effects. The slope is steeper in the social condition (which could be tested), claiming for a boost of RT in the social context.

Here the reviewer suggests to model RTs with lines, and to compare their slopes. However, in our experiments, RTs modulations in space do not follow a linear decrease. We actually observe a plateau for far distances, then boosts of RTs at critical closer distances.

-> However, the lack of effect on the left side makes the interpretation of the results difficult

A previous study demonstrated that lateral PPS is asymmetrical for right-handers, with a larger extent in the left side than the right side (Hobeika et al 2018). Considering this natural asymmetry of PPS extent for right-handed participants when doing the task alone, it is not surprising to find an anisotropy in the modulation of PPS in social contexts. We hypothesize that the left side being naturally larger, does not require to be further enlarged in social conditions (lines 431-442).

-> Furthermore, the boost effect is observed both sides in the competition condition.

We do not see this result as weakening the interpretation that PPS extent is sensitive to a high-level social instruction of collaboration. To the opposite, we think this provides evidence that inducing a competitive context does not impact PPS definition but just globally boosts reaction times, leaving the individual spatial definition of PPS unchanged.

-> Although the conclusions should be tempered, the fact that the effect was not observed on the left side could be related to the fact that the response was always delivered with the right hand. A control experiment with the response provided by the left hand could be valuable to better understand the articulation of the effects reported.

In the present study, participants received tactile simulations on one hand and responded with the other hand. Previous findings suggest that peri-hand PPS merges with peri-trunk PPS when hands are located near the body (Serino et al. 2015). Our aim was therefore to study peri-trunk PPS by applying tactile stimulation on the non-dominant hand, with both hands positioned in contact with the trunk. However, we cannot exclude the possibility that the present results are somehow associated to the peri-hand PPS of the non-dominant hand. An experiment using the left hand would not be the ideal control, as the task is based on speed-up effects and right-handers individuals are slower to respond with the left hand. However, following this remark, we have added more precision in the text, lines 121-125: “. In order to measure PPS of participants’ trunk, participants were instructed to keep their hands aligned with their mid-sagittal plane and in contact to their body [34]. With this instruction, the paradigm aimed at measuring PPS of participants’ trunk [34]. However, we can not exclude the possibility that our measure is somehow associated to the peri-hand PPS of the responding hand.”

-> It is also not clear why a side by side condition will produce an extension of PPS. Extension of PPS was obtained in a social context when two facing confederates performed a collaborative task (Teneggi et al. 2013; Coello et al., 2018). Perhaps, this could be argued.

Seating side by side a more neutral, less confrontational organization than face to face, that individuals spontaneously adopt to collaborate (Sommers R., 1965). Thus, this configuration is prone to highlighting a reorganization of the action space when individuals collaborate.

To our knowledge, there is no published study on PPS modulation in participants seated side by side. However, we describe in the introduction one experiment where the processing of stimuli in space is modulated for collaborating individuals seated side by side (second paragraph 2, lines 47-57). Following the comment of reviewer 1, we have also speculated more on why a side by side configuration would produce an extension of PPS (see discussion, paragraph 5 line 394-409 and paragraph 7, lines 419-430).

Minor points

Method

- What is the role of the visual target? How long was it displayed?

Because we wanted to avoid interactions between eye position and sound processing (e.g. Gruters et al 2017), we used a visual target to control for the gaze direction and visual stimulation of participants. The visual target was displayed during the whole duration of the experiment. Those information have been added in the manuscript lines 134-136.

« To control for the visual stimulation and gaze direction, participants were instructed to fix a permanent visual target located at 65cm in front of them »

- Concerning the conditions, there is 2 contexts (isolated, social), 2 directions (right, left) and 11 time delays. I thus wouldn't say that there is 22 conditions (but 44), with 10 trials in each. Also, the authors mention 22 conditions with 10 target stimuli in each, it is thus not clear why there is 440 trials.

Thank you for your attention. This mistake is now corrected. There are indeed 44 conditions.

- Concerning the seating condition, (left/right) it must be indicated if this condition was within/between subjects.

The seating condition is a between subject factor. We renamed this factor Partner Position. It is indicated lines 191-192

- Figure 1 shows an exact location for T1 to T9 according to the participant. How were these location determined (time to contact ?), were they perceived similarly by all participants (it is indicated that" The absolute distance estimation of sound sources is usually a difficult task for non-familiar stimuli in absence of reference, in which participants are not accurate). It would be interested to have an information concerning at which distance multisensory integration occurs for each participant.

The exact locations of the sound source in space for T1 to T9 correspond to the theoretical distances used to create the looming sound, based on physical laws of sound propagation.

The looming sound were synthesized using the Spat library of the Max/MSP(6.1.8) environment. The technique uses the Head-Related Transfer Function filters (or HRTF) that characterize the transformation of the signal during its propagation from a particular point in space to the entrance of the listener's ear canals. The HRTFs convey all the above described acoustical cues linked to interaural differences as well as to spectral cues caused by the diffraction on the listener's torso, head and pinnae. In addition to HRTFs, the reflections and the reverberation of the room are added, and the sound intensity of the direct sound is modulated to create a perception of moving sound in three dimensions.

The following table depicts the theoretical correspondences between time delays and distances from participants' body. As we point out in the manuscript, the absolute distance estimation of stationary sound sources is usually a difficult task, in which participants are not accurate for non-familiar sounds. However, participants are accurate in the comparison of distances between two sources at different distances. In our task, we are using looming sounds, and compare the effect of continuously varying sound distances. As distance perception in depth is accurate for relative comparison but not in term of absolute distance, we decided to present the results in terms of delay from the sound onset and not in terms of geometrical distances. We emphasized on this point in the manuscript to explain that we do not give a metric of PPS extent in space, as it could be misleading, but we added the theoretical range of distance in the third paragraph of the methods, line 157 - 158.

We agree that having information about participants' absolute distances estimation could be interesting. However, there is no efficient and reliable method that can be easily and rapidly used to test it. Nevertheless, we systematically checked that the sound was perceived as looming towards participants' body.

Delay	T1	T2	T3	T4	T5	T6	T7	T8	T9
Time from sound onset (ms)	0	375	750	1125	1500	1875	2250	2625	3000
Distance of sound (cm)	135	109.4	96.7	83.9	71.1	58.3	45.6	32.8	20

- Page 7. "RTs were significantly faster when the tactile stimulus occurred at T4 compared to when the tactile stimulus occurred at T5". This is counterintuitive and seems the opposite to what is reported Figure 2.

Thank you again for your attentive reading. This was indeed a mistake. RTs were actually faster when the tactile occurred at **T5** compared to **T4**. We corrected this in the manuscript.